Molecular Biology and Physiology

# Single-Cell RNA Sequencing Reveals that the Switching of the Transcriptional Profiles of Cysteine-Related Genes Alters the Virulence of *Entamoeba histolytica*

Meng Feng,[a] Yuhan Zhang,[a] Hang Zhou,[a] Xia Li,[a] Yongfeng Fu,[a] ⬤ Hiroshi Tachibana,[b] ⬤ Xunjia Cheng[a,b]

[a]Department of Medical Microbiology and Parasitology, School of Basic Medical Sciences, Fudan University, Shanghai, China
[b]Department of Infectious Diseases, Tokai University School of Medicine, Isehara, Kanagawa, Japan

Meng Feng and Yuhan Zhang contributed equally to this work. Author order was determined both alphabetically and in order of increasing seniority.

**ABSTRACT** *Entamoeba histolytica* is an intestinal protozoan that causes human amoebic colitis and extraintestinal abscesses. Virulence variation is observed in the pathogenicity of *E. histolytica* trophozoites, but the detailed mechanism remains unclear. Here, a single trophozoite was cultured alone, and the progeny of the trophozoites of each generation were subjected to single-cell RNA sequencing (scRNA-seq) to study the transcriptional profiles of trophozoites. The scRNA-seq analysis indicated the importance of sulfur metabolism and the proteasome pathway in pathogenicity, whereas the isobaric tags for relative and absolute quantitation (iTRAQ) proteomic analysis did not identify the bulk trophozoites. The trophozoite improved the synthesis of cysteine under cysteine-deficient conditions but downregulated the expression of the intermediate subunit of the lectin of *E. histolytica* trophozoites and retained the expression of the heavy subunit of lectin, resulting in decreased amoebic phagocytosis and cytotoxicity. The variation in the transmembrane kinase gene family might be critical in regulating the proteasome pathway. Thus, the scRNA-seq technique provided an improved understanding of the biological characteristics and the mechanism of virulence variation of amoebic trophozoites.

**IMPORTANCE** Studies on the trophozoite of *Entamoeba histolytica* suggested this organism could accumulate polyploid cells in its proliferative phase and differentiate its cell cycle from that of other eukaryotes. Therefore, a single-cell sequencing technique was used to study the switching of the RNA transcription profiles of single amoebic trophozoites. We separated individual trophozoites from axenic cultured trophozoites, CHO cell-incubated trophozoites, and *in vivo* trophozoites. We found important changes in the sulfur and cysteine metabolism in pathogenicity. The trophozoites strategically regulated the expression of the cysteine-rich protein-encoding genes under cysteine-deficient conditions, thereby decreasing amoebic phagocytosis and cytotoxicity. The single-cell sequencing technique shows evident advantages in comparison with the isobaric tags for relative and absolute quantitation (iTRAQ) proteomic technology (bulk trophozoite level) and reveals the regulation strategy of trophozoites in the absence of exogenous cysteine. This regulation strategy may be the mechanism of virulence variation of amoebic trophozoites.

**KEYWORDS** *Entamoeba histolytica*, single-cell RNA sequencing, cysteine-related gene, virulence

The amoebiasis caused by infection with *Entamoeba histolytica* is one of the most important protozoan infection, which is prevalent all over the world. Fifty million people are estimated to be infected with amoebic colitis or extraintestinal abscesses,

Address correspondence to Hiroshi Tachibana, htachiba@is.icc.u-tokai.ac.jp, or Xunjia Cheng, xjcheng@shmu.edu.cn.

resulting in 40,000 to 100,000 deaths annually (1–5). Owing to the serious pathogenicity of *E. histolytica*, the virulence variation of *E. histolytica* trophozoites has attracted much attention in research. The virulence of trophozoites decreases during long-term culture *in vitro* (6, 7). Simultaneously, previous studies have suggested that the virulence of *E. histolytica* trophozoites increases during pathogenicity *in vivo* (8, 9). However, the detailed mechanism remains unclear. The variations in proteome, transcriptome, and regulatory mechanisms in the process of virulence variation of trophozoites need to be studied further.

Cysteine is one of most important nutrient elements of *E. histolytica* (10). The cysteine synthesis pathway is one of several amino acid synthesis pathways retained by amoeba (11), and many cysteine-rich proteins of *E. histolytica* play important roles. The CXXC motif-rich intermediate subunit (Igl) of galactose- and *N*-acetyl-D-galactosamine (Gal/GalNAc)-inhibitable lectin of *E. histolytica* contributes to adherence (12–14). Igl-1 and Igl-2 are the two isoforms of Igl (15), and Igl-1 seems to be closely associated with the pathogenicity of *E. histolytica* (16). Moreover, *E. histolytica* has an estimated 90 transmembrane kinases (TMKs), which were previously identified as a CXXC motif-rich tyrosine protein kinase family. These TMKs form a large family with highly variable extracellular domains homologous to Igl (17–20). TMKs regulate diverse cellular processes in animals, including cell survival, cell proliferation, biological metabolism, and migration (21). The presence of multiple receptor kinases in the plasma membrane offers a potential explanation of the ability of the parasite to respond to the changing environment of the host. Thus, TMKs play critical roles as signal perceivers and transducers in higher eukaryotes (22, 23) and may be important in the regulation of the transcriptional profiles of amoeba.

Next-generation sequencing of mRNAs in single cells is a major advancement (24) in studying the switching of transcriptional profiles of cysteine-related genes. Transcriptome sequencing at a single-cell level enables the discovery of covert cell-specific changes caused by various stimuli in the transcriptome (25–27). Individual cells are the basic building blocks of organisms, and each cell is unique. Performing bulk RNA sequencing often masks such uniqueness and fails to reveal latent changes. Single-cell RNA sequencing (scRNA-seq) has emerged as a revolutionary tool to uncover the uniqueness of each cell, thus enabling the study of biology at microscopic resolution and addressing questions that could not be answered previously (28–30). This technology is particularly suitable for the study of the amoeba transcriptome because amoebic trophozoites are single-cell protozoa, which act independently of their biological effects.

In the present study, we have used scRNA-seq to quantify the levels of trophozoite mRNAs *in vitro* and *in vivo* and clarify the variation and differences in the transcriptome. This strategy is beneficial in understanding the variation in the transcriptional profiles and the function and regulation of cysteine metabolism and cysteine-related proteins.

## RESULTS

**Single-cell RNA sequencing and identification of trophozoite expression profiles.** Single trophozoites were tested for the switching of transcriptional profiles through incubation with CHO cells *in vitro* or inoculation into hamster livers (Fig. 1). A total of 17, 21, and 7 single trophozoites from axenic culture, CHO incubation, and *in vivo* groups, respectively, were successfully subjected to scRNA-seq (Fig. 2A). The differential expression analysis of the single trophozoites from these three groups suggested that the trophozoites from the CHO incubation group had a higher differentially expressed gene level than those from the other two groups. The average fold change of differentially expressed genes in the CHO incubation group was 20% higher, indicating an active transcriptional state when trophozoites were coincubated with CHO cells *in vitro* (Fig. 2B).

A single-cell principal-component analysis (PCA) plot further showed the distinctions among the three single-trophozoite groups. The expression profiles of the axenic culture and the *in vivo* groups had minimal differences. However, minimal overlap was

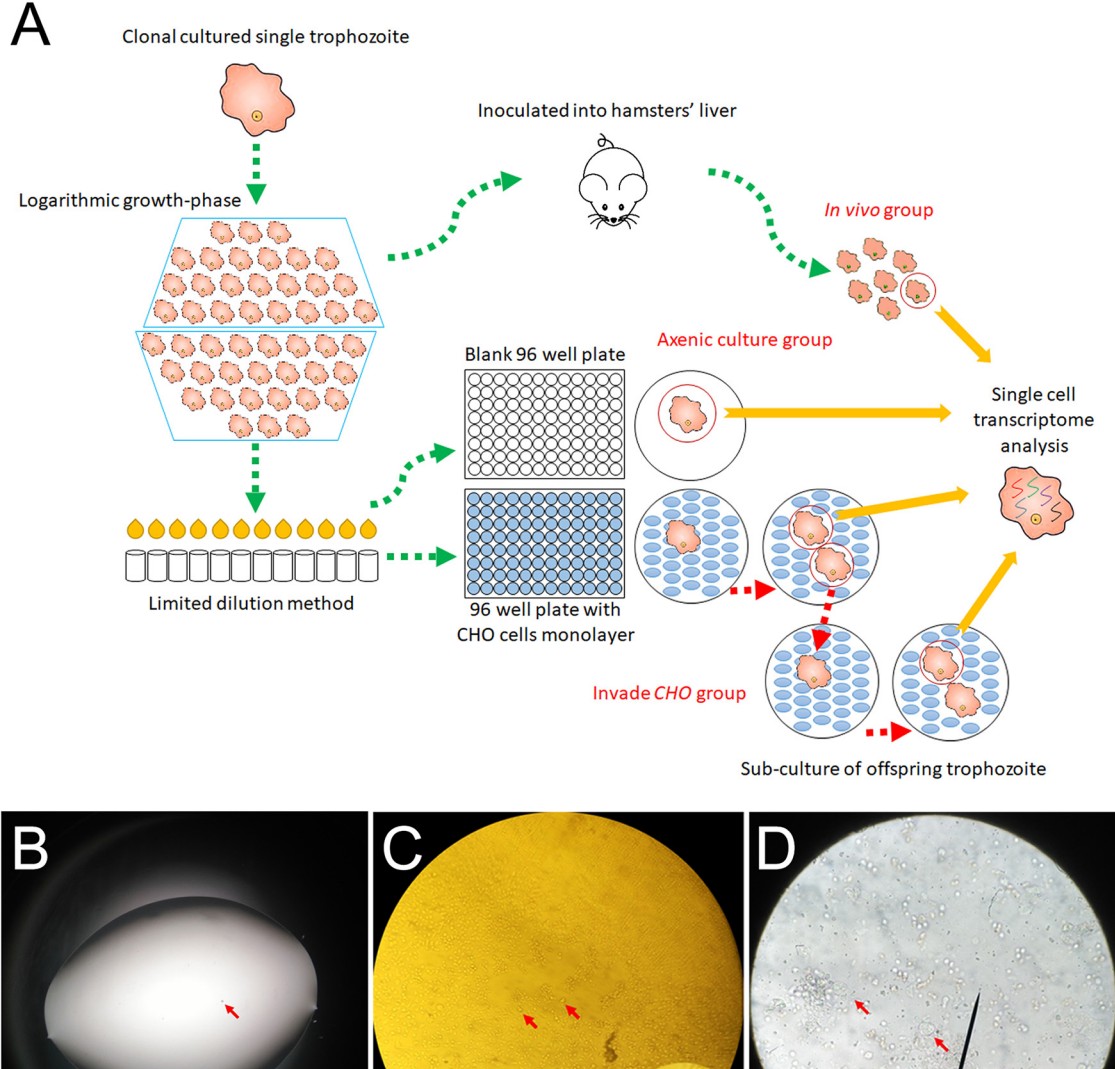

**FIG 1** Technological process of capturing single trophozoites for scRNA-seq. (A) Three groups of single trophozoites are captured using a limited dilution method and subjected to scRNA-seq. (B) Single trophozoite (arrow) in 1 μl culture medium. (C) Asexually proliferated trophozoites (arrows) in the CHO cell monocyte layer. (D) Trophozoites (arrows) in liver abscess tissue.

observed in the expression profile of the CHO incubation group with those of the two other groups (Fig. 2C). The trophozoites from the CHO incubation group were discrete, which suggested high variability of the expression profiles in the CHO incubation group, and the minimal difference of the *in vivo* group from the axenic culture group indicated that a few key variations in the genes in the expression profiles probably determined the pathogenicity of trophozoites *in vivo*.

We clustered all single-trophozoite profiles as a heat map on the basis of different groups to gain insights into the extent of this diversity. First, we confirmed that the single trophozoites from the CHO incubation group displayed an upregulation of genes that were different from those for the two other groups (region I), and the single trophozoites from the *in vivo* (region II) or the axenic culture (region III) group displayed separate upregulations of genes (Fig. 3A). Each group had its own gene upregulation profile, suggesting that the switching of transcriptional profiles after incubating with CHO cells *in vitro* or inoculation into hamster liver was quite different. The genes that were

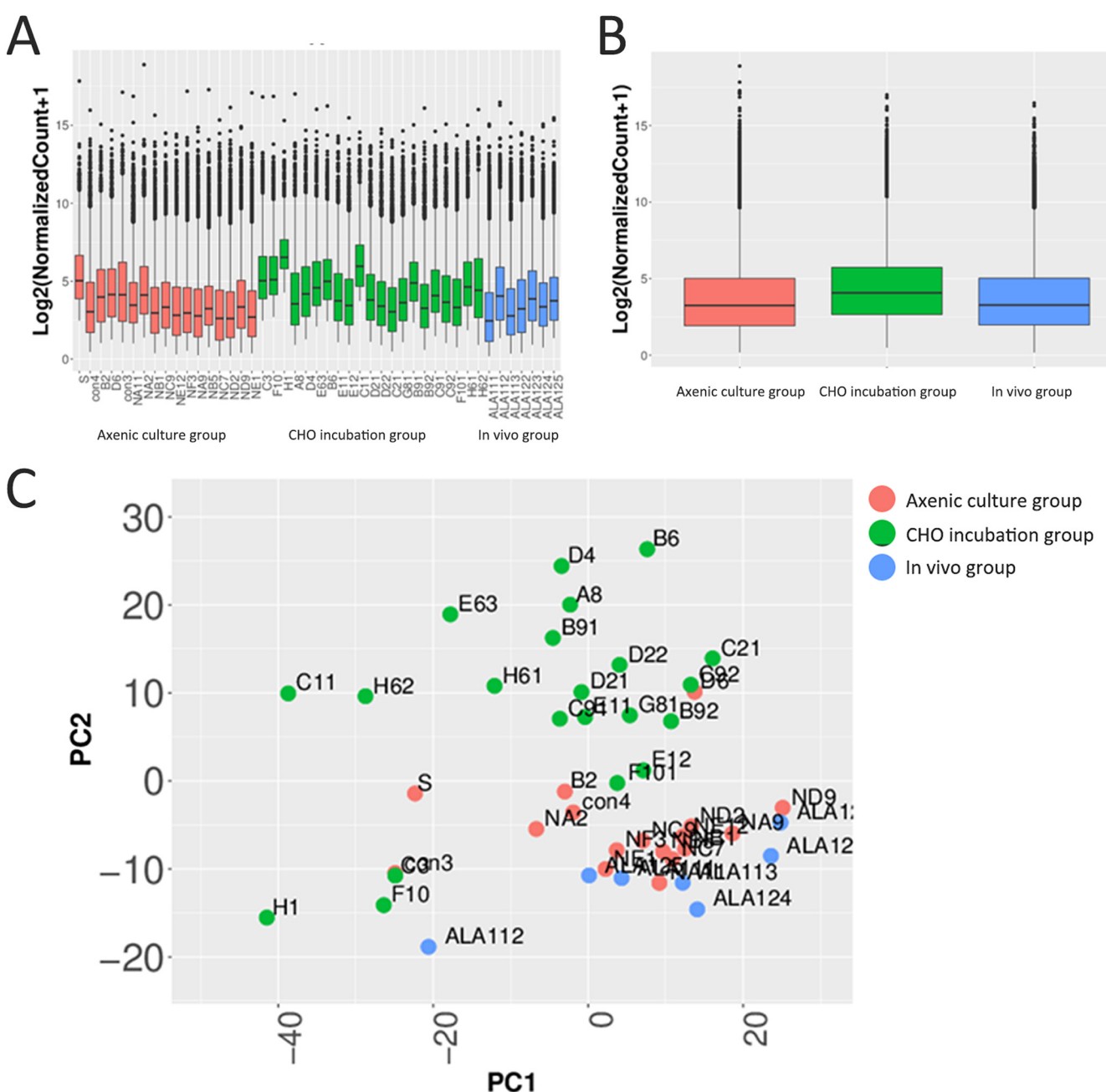

**FIG 2** Evident changes in the expression profiles of CHO-incubated trophozoites in scRNA-seq analysis. (A) Differentially expressed gene levels in individual trophozoites in scRNA-seq analysis. (B) Average differentially expressed gene levels for three groups in scRNA-seq analysis. (C) Principal-component analysis of differential groups of single trophozoites by using all differentially expressed genes. Single trophozoites of axenic culture, CHO incubation, and *in vivo* groups are marked in red, green, and blue, respectively.

statistically enriched or reduced were identified to characterize the scRNA-seq trailblazer transcriptional signature. Volcano plots showed 304 upregulated and 359 downregulated genes (CHO incubation group versus axenic culture group) (Fig. 3B), 431 upregulated and 403 downregulated genes (*in vivo* group versus axenic culture group) (Fig. 3C), and 391 upregulated and 474 downregulated genes (*in vivo* group versus CHO incubation group) (Fig. 3D). The results suggested a slightly evident downregulation of gene expression in the CHO incubation group.

**Heterogeneous regulation of gene expression across single trophozoites *in vitro* and *in vivo*.** The genes of the expression profiles were divided into three groups, namely, biological process (BP), cell component (CC), and molecular function (MF). To

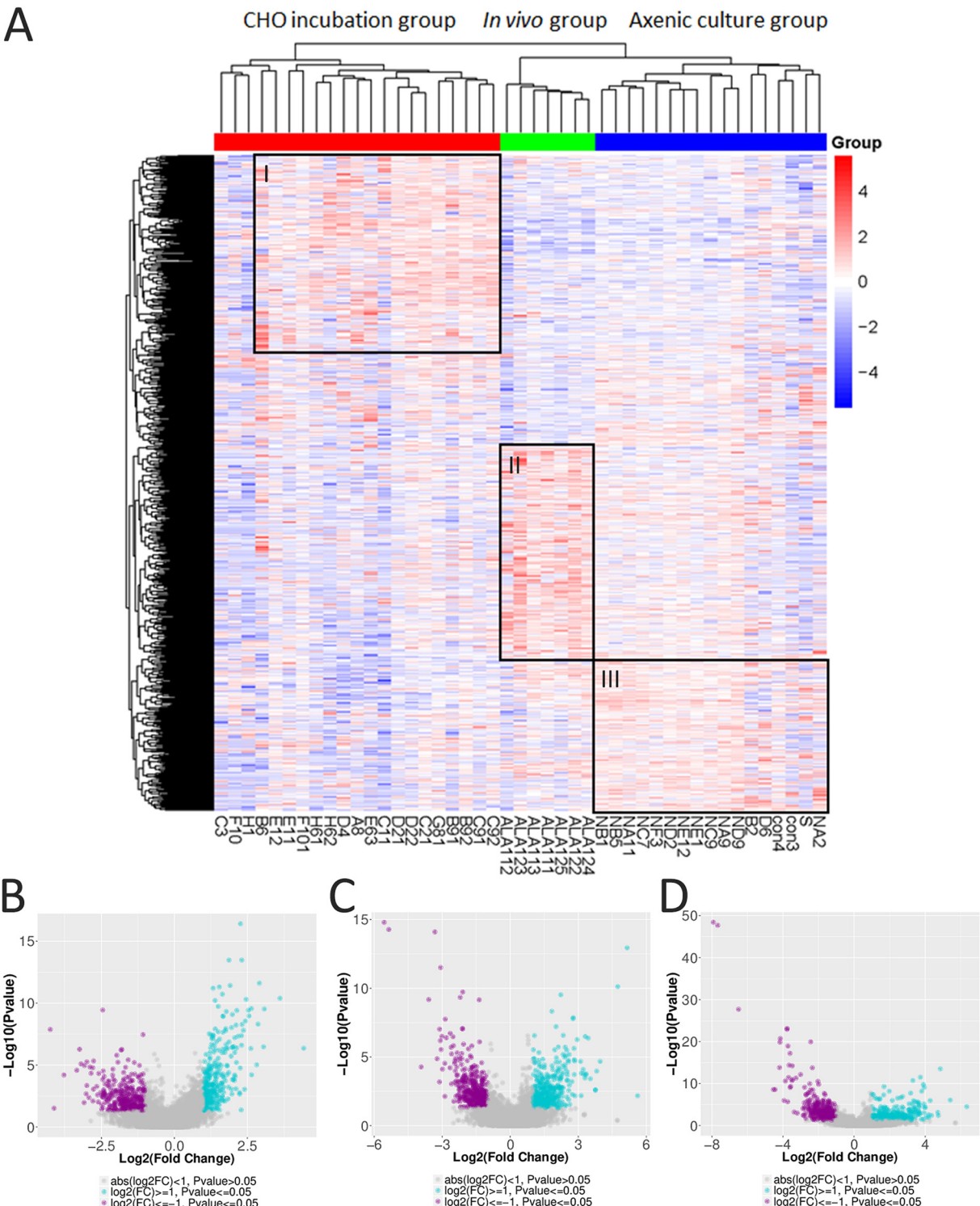

**FIG 3** Transcriptional variation in single trophozoites from three groups in scRNA-seq analysis. (A) Heat map of differentially expressed genes in single trophozoites of CHO incubation (red), *in vivo* (green), and axenic culture (blue) groups. Black boxes contain gene clusters expressed at significantly higher levels in CHO incubation (box I), *in vivo* (box II), and axenic culture (box III) groups. Volcano plots of expression fold change in CHO incubation group versus axenic culture group (B), *in vivo* group versus axenic culture group (C), and *in vivo* group versus CHO incubation group (D). (B to D) Upregulated genes are shown in blue, whereas downregulated genes are shown in purple.

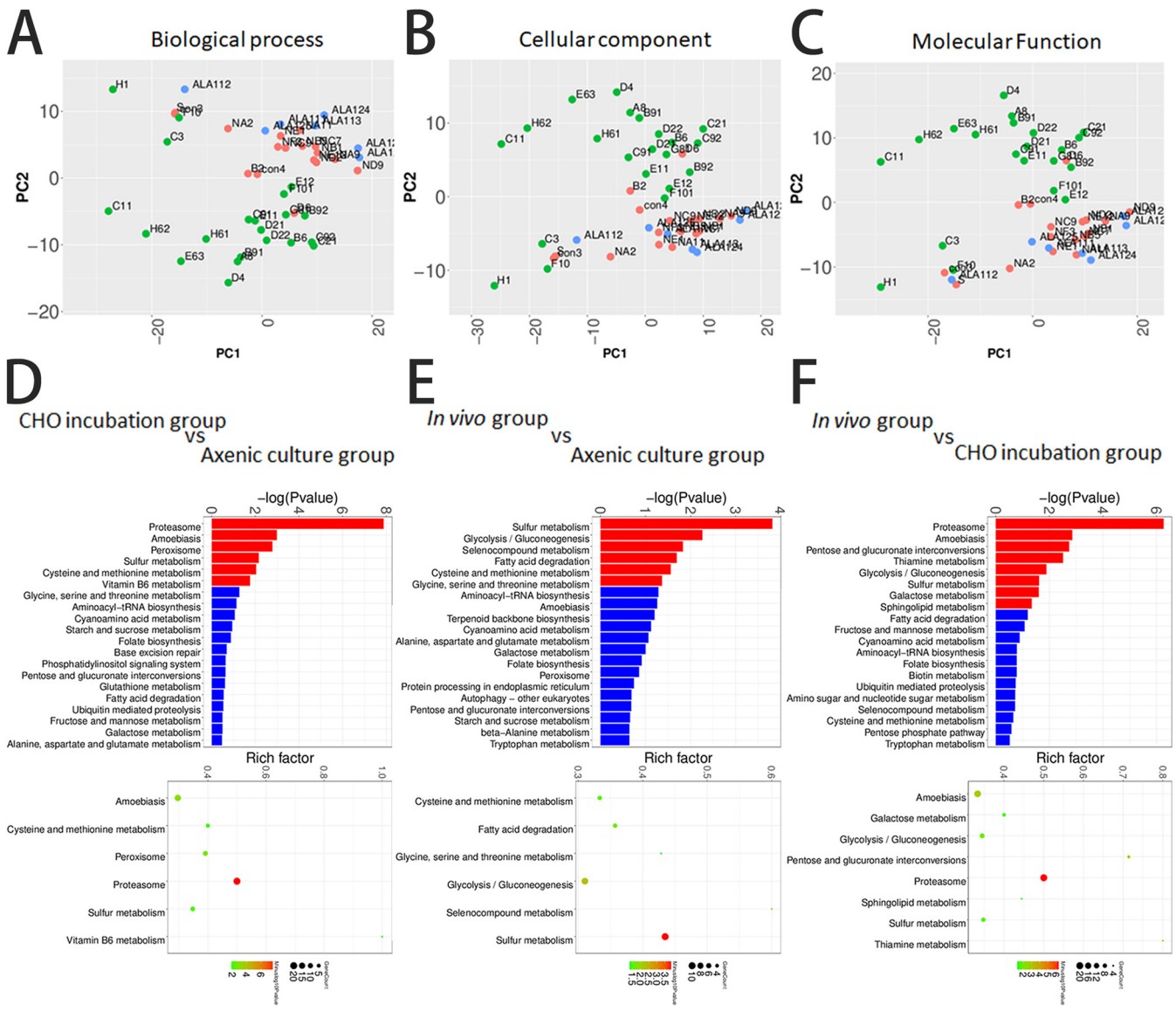

**FIG 4** KEGG pathway classification of transcriptional variation in single trophozoites from three groups in scRNA-seq analysis. Principal-component analysis of single trophozoites by using differentially expressed genes classified for biological process (A), cellular component (B), and molecular function (C). Single trophozoites of axenic culture, CHO incubation, and *in vivo* groups are marked in red, green, and blue, respectively. Condensed gene ontology analysis of upregulated and downregulated genes comparing CHO incubation group versus axenic culture group (D), *in vivo* group versus axenic culture group (E), and *in vivo* group versus CHO incubation group (F). (D to F) Bar charts displaying upregulated (red) or downregulated (blue) genes in the comparison group.

determine the key variation in the gene expression profile within the three components, we clustered all single-trophozoite profiles as single-cell PCA plots by using separated BP, CC, and MF genes (see Table S3 in the supplemental material). The expression profiles of the axenic culture and the *in vivo* groups had minimal differences in the CC and the MF genes and a slight difference in the BP genes, indicating that the key variation in the gene expression profile *in vivo* was because of the BP genes. Conversely, the high-expression profiles of CC and MF and the low-expression profile of BP were found in the CHO incubation group (Fig. 4A to C). This finding suggested a quite different variability in the expression profile of the CHO incubation group and a difference in survival or pathogenicity by incubating trophozoites with CHO cells *in vitro* and inoculating trophozoites into hamster liver.

Further transcriptional signatures were analyzed using the Kyoto Encyclopedia of Genes and Genomes (KEGG) enrichment, and the statistically enriched KEGG pathways

were identified. The pathways of proteasome, amoebiasis, peroxisome, and sulfur metabolism were upregulated (CHO incubation group versus axenic culture group). The pathways of sulfur metabolism, glycolysis/gluconeogenesis, selenocompound metabolism, and fatty acid degradation were upregulated (*in vivo* group versus axenic culture group). The pathways of proteasome, amoebiasis, pentose and glucoronate interconversions, and thiamine metabolism were upregulated (*in vivo* group versus CHO incubation group) (Fig. 4D to F). The levels of the key enzymes of sulfur metabolism (i.e., sulfate adenylyltransferase and methionine gamma-lyase [MGL]) increased, and the levels of the key enzymes of the cysteine synthase (CS) genes decreased in most single trophozoites from the *in vivo* group. *E. histolytica* had an incomplete pathway of sulfur metabolism (see Fig. S1 in the supplemental material), thus suggesting that CS and MGL were also related to cysteine metabolism and could play an important role in the survival or pathogenicity of trophozoites.

**Related regulation of CS and Igl during each intrageneration.** A high expression level of CS and low expression levels of MGL, Igl-1, and Igl-2 were identified in the CHO incubation group. MGL was only highly expressed in the *in vivo* group (Fig. 5A). A cysteine-deficient medium was used to culture trophozoites for 24, 48, and 60 h and to study the expression relationship between cysteine and lectin. Quantitative real-time RT-PCR (qRT-PCR) was used to confirm the expression patterns of CS and lectin genes. The expression level of CS remained high under cysteine-deficient culture conditions at 48 and 60 h, but the expression levels of MGL, Igl-1, and Igl-2 decreased under the cysteine-deficient conditions at 48 and 60 h; the expression of the heavy subunit of lectin (Hgl) was retained (Fig. 5B). Moreover, the expression levels of lectin and CS genes were compared using each single trophozoite. Results indicated significantly positive correlations between CS and Igl-1 and between CS and Igl-2 (Fig. 6 and 7).

F6-7

**Decrease in phagocytosis and cytotoxicity in cysteine-deprived trophozoites.** The expression levels of cysteine proteinase and amoebapore (AP) genes slightly decreased in the CHO incubation group. In addition, cysteine protease 2 (CP2), CP5, and AP-A genes were slightly decreased under cysteine-deficient conditions at 60 h (Fig. 5B). Further phagocytic and cytotoxic assays were used to evaluate the virulence changes in the trophozoites under cysteine-deprived conditions. Cysteine-deprived trophozoites were labeled with carboxyfluorescein succinimidyl ester (CFSE) and coincubated with DiD-labeled heat-killed Jurkat cells. All trophozoites and cells were fixed, and phagocytosis was assessed using flow cytometry (Fig. 8A). Cysteine deficiency inhibited the phagocytosis of trophozoites, and the percentages of trophozoites that ingested Jurkat cells decreased to 75.9% and 78.6% at 5 and 10 min, respectively. In the control group, the percentages were 90.1% and 91.4% at 5 and 10 min, respectively. In the cytotoxic assay, cysteine-deprived trophozoites were coincubated with Jurkat cells, and after propidium iodide (PI) staining, dramatically reduced amoebic cytotoxic activity was observed. The percentage of the killed Jurkat cells decreased from 10 min (10.4%) to 20 min (13.3%), whereas the percentage of killed Jurkat cells increased to 26.1% (10 min) and 49.5% (20 min) in control amoeba (Fig. 8B and Fig. S2). Results indicated that cysteine deficiency impaired the virulence of trophozoites by significantly reducing the expression levels of cysteine-rich genes (Igls) and slightly decreasing CP and AP.

**Latent transcriptional profiles identified at the single-cell level compared to that at the bulk level.** The isobaric tags for relative and absolute quantitation (iTRAQ) proteomic analysis revealed that 603 of the 2,471 proteins, including 430 upregulated and 173 downregulated proteins, were identified to be differentially expressed by a fold change cutoff ratio of >1.2 or <0.833 after trophozoites were incubated with CHO cells for 2 h. When trophozoites were incubated with CHO cells for 4 h, 609 proteins, including 400 upregulated and 209 downregulated proteins, were differentially expressed. All differentially expressed proteins in the group including trophozoites incubated with CHO cells for 2 h were categorized using the Gene Ontology (GO) analysis based on the international standardized gene functional classification system. The differentially expressed proteins were found to be involved in BP (e.g., cellular process,

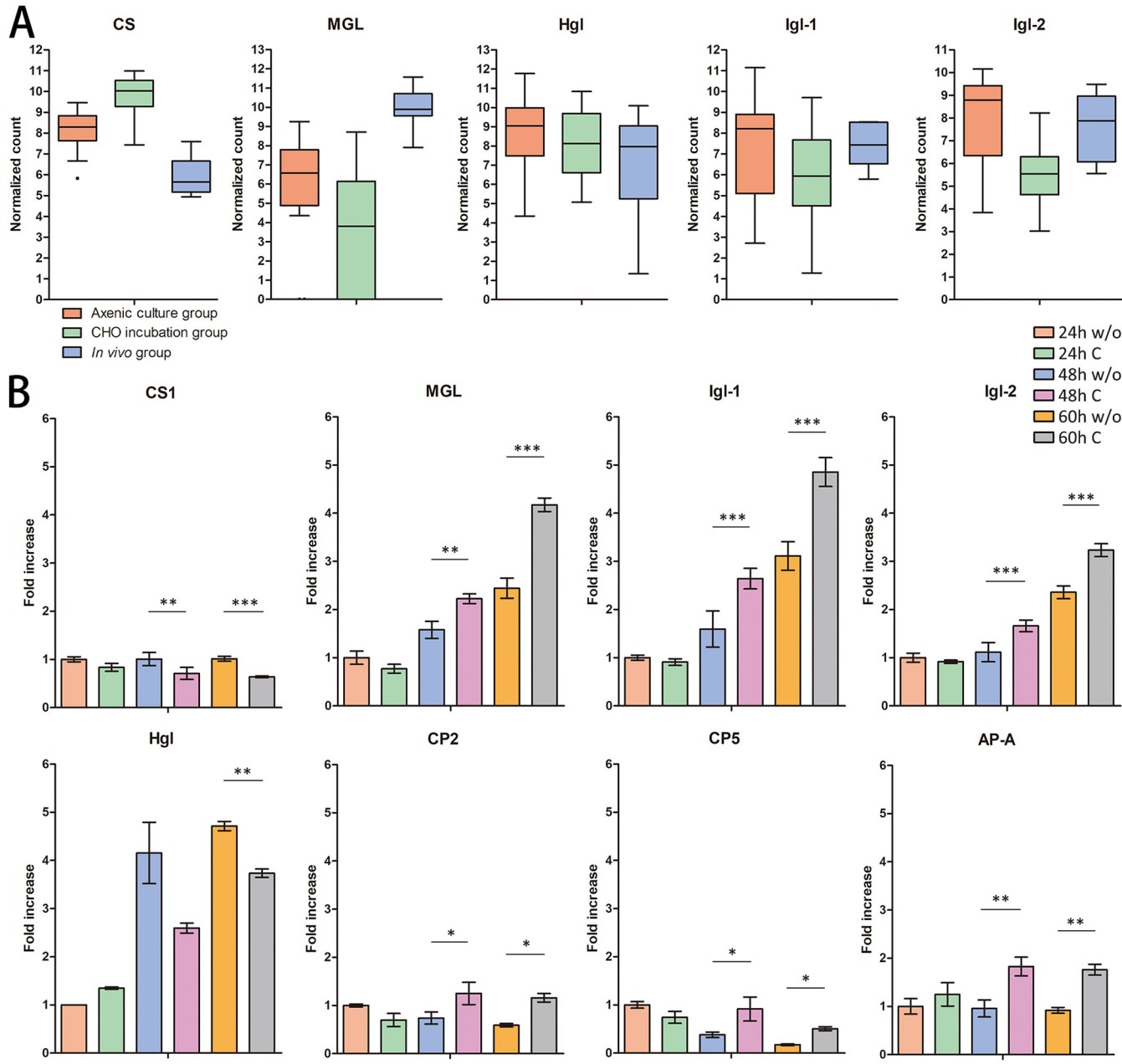

**FIG 5** Gene expression in scRNA-seq and quantitative real-time PCR. (A) Differential gene expression in scRNA-seq. Single trophozoites of axenic culture, CHO incubation, and *in vivo* groups are marked in orange, green, and blue, respectively. (B) qRT-PCR assays of CS, MGL, Hgl, Igl-1, Igl-2, CP2, CP5, and AP-A genes of trophozoites cultured under normal or cysteine-deficient conditions. c, medium contained normal concentration of cysteine; w/o, medium without cysteine. The gene expression levels are represented using the $2^{-\Delta\Delta CT}$ of the target gene relative to the $\beta$-actin gene. The y axes are linear coordinates, and numbers correspond to the fold increase over 1.0 given to the axenic culture group. *, $P < 0.05$; **, $P < 0.01$; ***, $P < 0.001$.

metabolic process, and biological regulation) (see Fig. S3A), CC (e.g., cell, cell part, and organelle) (Fig. S3B), and MF (e.g., binding and catalytic activity) (Fig. S3C).

The iTRAQ proteomic analysis detected 15 types of TMKs, 3 types of Hgl, 2 types of Igl, 3 types of AP, and 4 types of cysteine proteinases in all four groups. Among them, the expressions of TMK3, TMK6, TMK37, TMK59, TMK60, TMK71, Hgl, Igl, cysteine proteinase 1, and cysteine proteinase 5 increased after the trophozoites were incubated with CHO cells. Alternatively, the expression of genes in the proteasome pathway did not significantly change (see Tables S1 and S2). Results suggested that TMKs and virulence-associated proteins were routinely expressed in bulk trophozoites under *in vitro* culture conditions.

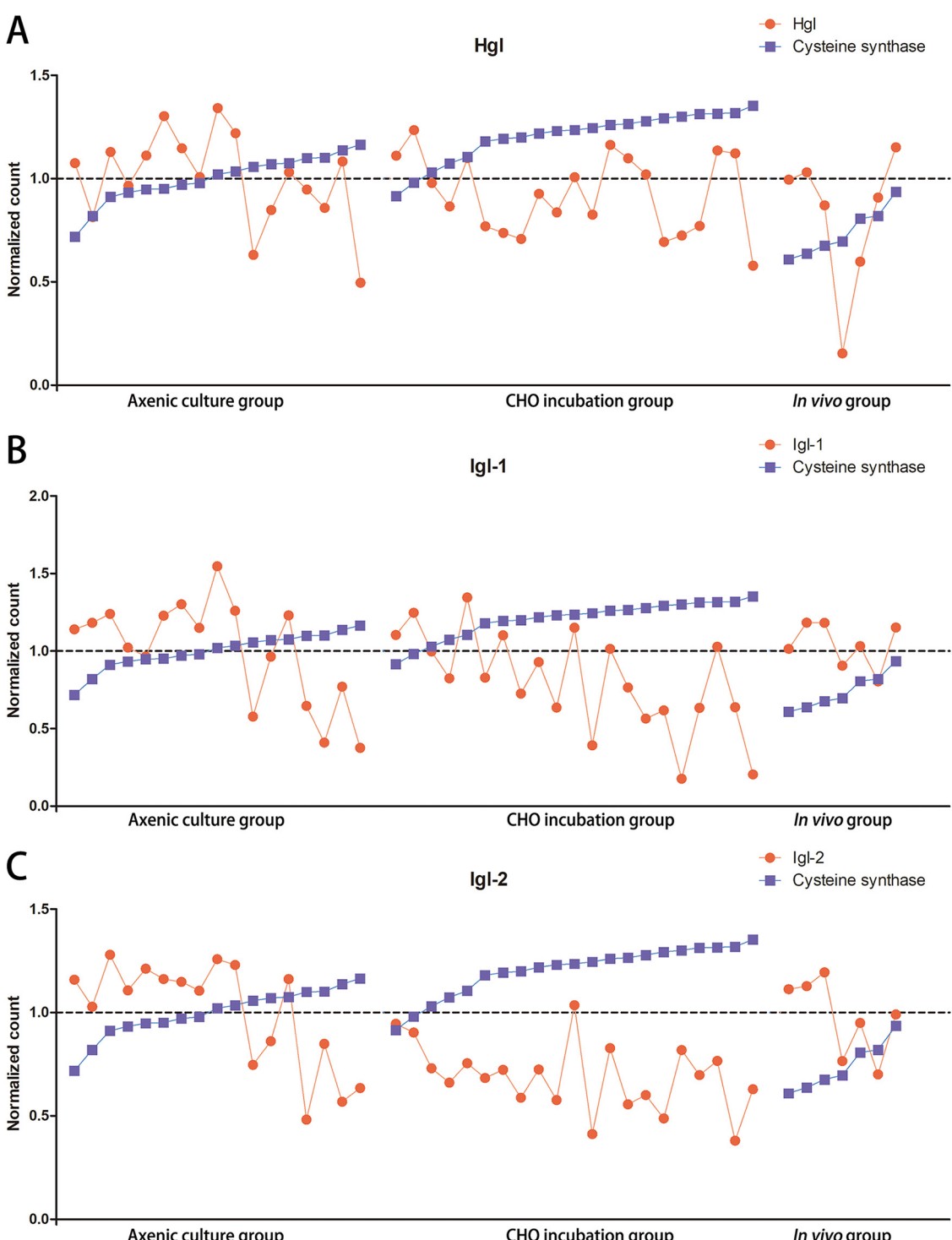

**FIG 6** Expression levels of lectin and CS genes compared using single trophozoites. Related regulation between Hgl and CS (A), Igl-1 and CS (B), and Igl-2 and CS (C).

In scRNA-seq, the higher expression of genes in the proteasome pathway in trophozoites from the CHO incubation group (e.g., Rpn3, Rpn5, Rpn6, Rpn9, Rpn11, Rpt1, Rpt3, Rpt4, Rpt6, and Rpt13) suggested a different condition of pathogenicity *in vitro*. However, the iTRAQ proteomic analysis did not identify an evident increase in the proteasome pathway in bulk trophozoites after incubation with CHO cells, indicating that

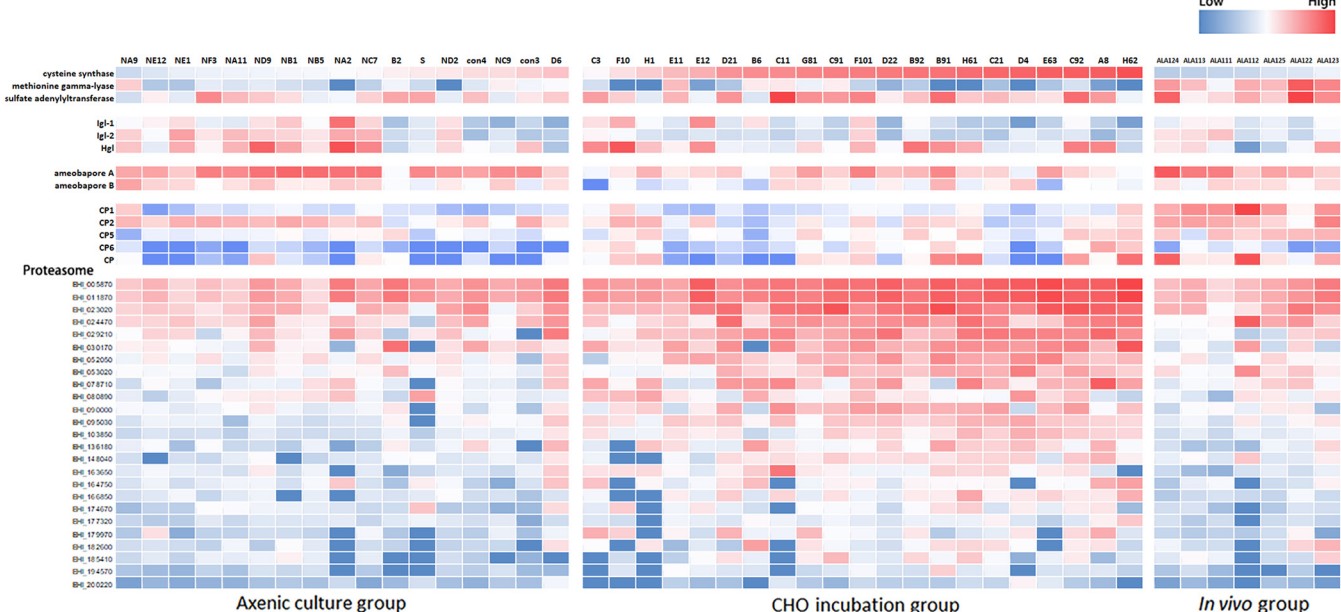

**FIG 7** Heat map of cysteine-related genes, CP, AP, and proteasome genes in single trophozoites of axenic culture, CHO incubation, and *in vivo* groups. The expression of cysteine synthase in each group was used for ordering.

transcriptome sequencing at a single-cell level enabled the discovery of covert trophozoite-specific changes in pathogenicity (Table S2).

**Selective transcription and coregulated TMK genes sharing BP.** Transmembrane protein kinases are important protein kinase families regulating the metabolism of trophozoites. The TMK gene expression profiles from bulk trophozoites and reverse-transcription PCR (primers listed in Table S4) indicated that most TMK genes were expressed during *in vitro* culture (Table S1). However, single-trophozoite sequencing revealed the unique expression profiles of TMK genes. The TMK genes selectively transcribed at the single-trophozoite level suggested that analysis at the bulk level would conceal the real transcriptional profiles (see Table S5). The selective expression of TMKs was particularly evident under cysteine-deficient conditions in the CHO incubation group, with increasing high transcription of TMK genes and decreasing low transcription of TMK genes (see Fig. S4A). In addition, the differential gene expression was evident for several TMKs. The high expression levels of TMK65 and TMK87 were identified in the CHO incubation and the *in vivo* groups. TMK29, TMK35, TMK37, TMK94, and TMK96 were only highly expressed in the CHO incubation group. TMK3 and TMK63 were only highly expressed in the *in vivo* group (Fig. S4B). Single-trophozoite TMK gene expression profiles suggested that single-cell-level analysis allowed an accurate observation of gene expression profiles.

**Switching of TMK gene transcriptional profiles regulating the proteasome pathway in single trophozoites.** Single-trophozoite profiles were clustered as a heat map on the basis of the TMK and the proteasome pathway genes to gain insights into the differences in the single-trophozoite expression profiles *in vitro* and *in vivo*. We confirmed that the single trophozoites from the CHO incubation group displayed an upregulation of proteasome and TMK genes distinct from those for the two other groups (Fig. S4C and Table S4). Moreover, the transcription levels of TMK genes were positively correlated with that of CS (Fig. 9). Results suggested that the upregulation of TMK genes was related to increases in proteasomes, and TMKs might play important roles in regulating the transcriptional profiles of trophozoites interacting with other organisms and under cysteine-deficient conditions.

After the single trophozoites were added to the CHO cell monocyte layers, multiple offspring trophozoites from the original trophozoites were collected. Trophozoites

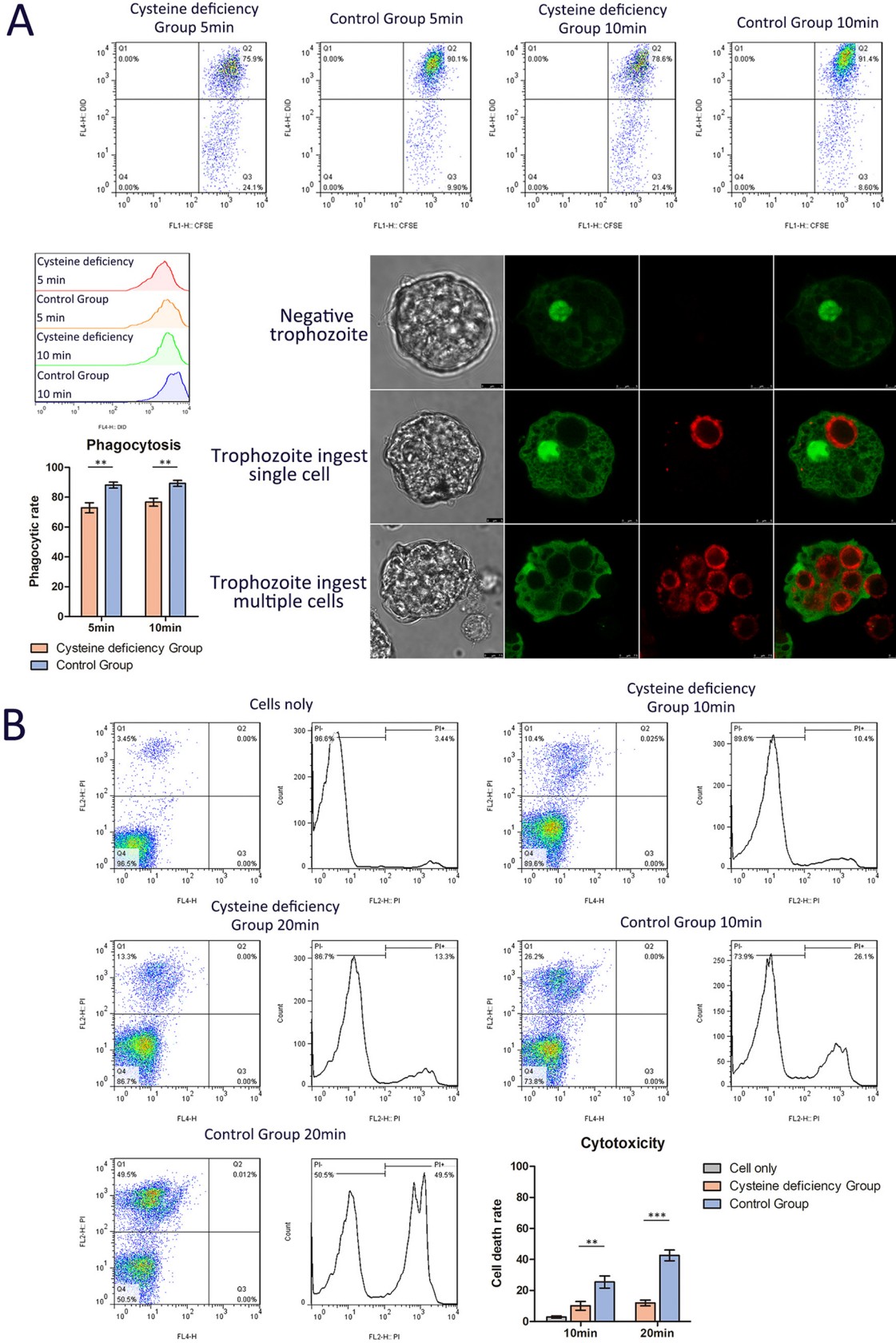

**FIG 8** Flow cytometry and fluorescence imaging of trophozoites and cells in amoebic phagocytic and cytotoxic assays. (A) Cysteine deficiency decreases amoebic phagocytosis. Normal or cysteine-deprived trophozoites were coincubated with DiD-labeled heat-killed

proliferated up to four generations (F1 to F4). In the single-cell PCA plot of these three distinct single-trophozoite groups, the trophozoites from the CHO incubation group were discrete and had highly variable expression profiles. However, the transcriptomes of trophozoites changed a lot among different batches and a little within the same batches (see Fig. S5). A heat map based on the TMK pathway genes presented the unique expression profiles of single trophozoites. The gene expression profiles of the same offspring were similar (H1, C3, and F10; H61 and H62), whereas the expression profiles of offspring and mother generations were different (B6 to E11 and E12; C11 to B91 and B92) or similar (C21, D21, D22, C91, C92, and F101) (Fig. 9). Results suggested the high variability of unique TMK gene expression profiles in the trophozoite proliferation process, which might be because the trophozoites that proliferated *in vitro* could not remain virulent compared with the parasites *in vivo*. Additionally, no significant regularity of periodic variation was observed in the single trophozoites from the axenic culture and the *in vivo* groups, because an accurate determination of cell replication cycles in these two groups was not possible (Fig. S4C).

## DISCUSSION

The scRNA-seq technique can uncover the uniqueness of each cell and address the questions of microstructural transcription variations that were not able to be answered previously (31–36). An scRNA-seq study on malarial parasites involving the measurement of gene expression in thousands of individual parasites has helped address key questions that involve small subpopulations of parasites and revealed a signature of sexual commitment in malarial parasites (37). Considering that amoebic trophozoites are unicellular organisms, sequencing mRNAs at a single-trophozoite level enables the discovery of cell-specific changes caused by intrinsic or extrinsic stimuli in the transcriptome. This study has compared the differential expression of proteins between bulk and single trophozoites after incubation with CHO cells. The scRNA-seq identified a proteasome pathway that is upregulated in the CHO incubation group (Fig. 7), whereas the iTRAQ proteomic analysis did not identify such a large increase in the proteasome pathway in bulk trophozoites after incubation with CHO cells (see Table S2 in the supplemental material). Results suggest that performing bulk RNA sequencing often masks such uniqueness and fails to reveal latent changes. Only a portion of trophozoites activated the pathways of proteasome and other transcript proteins, indicating that the biological characteristics of trophozoites are inconsistent or asynchronous *in vitro*.

This study demonstrates that the expression profile of the CHO incubation group has very minimal overlap with that of the *in vivo* group. The trophozoites from the CHO incubation group display high variability of expression profiles, in which the pathways of proteasome and amoebiasis are upregulated. However, the pathways of sulfur metabolism and glycolysis/gluconeogenesis were upregulated in the *in vivo* group (Fig. 4). A previous study indicated that the decrease in the virulence of *E. histolytica* during prolonged periods in the axenic culture is, in part, because of their increased susceptibility to the amoebicidal effects of macrophages (6). Recent studies have demonstrated that the origins, benefits, and triggers of amoebic virulence are complex (9, 38, 39). Amoebic pathogenesis entails the depletion of the host mucosal barrier, adherence to the colonic lumen, cytotoxicity, and invasion of the colonic epithelium (40, 41). The host and the parasite genotypes influence the development of disease, as they govern the regulatory responses at the host-pathogen interface (38, 39, 41). Host environmental factors determine parasite transmission and shape the colonic microenvironment that *E. histolytica* infects (42–45). When the trophozoites of *E. histolytica* cope with environmental stress *in vitro*, many genes have undergone disordered changes. *In*

**FIG 8** Legend (Continued)
Jurkat cells at 37°C for 5 or 10 min. Representative images are shown. (B) Cysteine deficiency decreases amoebic cytotoxicity. Normal or cysteine-deprived trophozoites were coincubated with living Jurkat cells at 37°C for 10 or 20 min. Cells were stained with PI. **, $P <$ 0.01; ***, $P <$ 0.001.

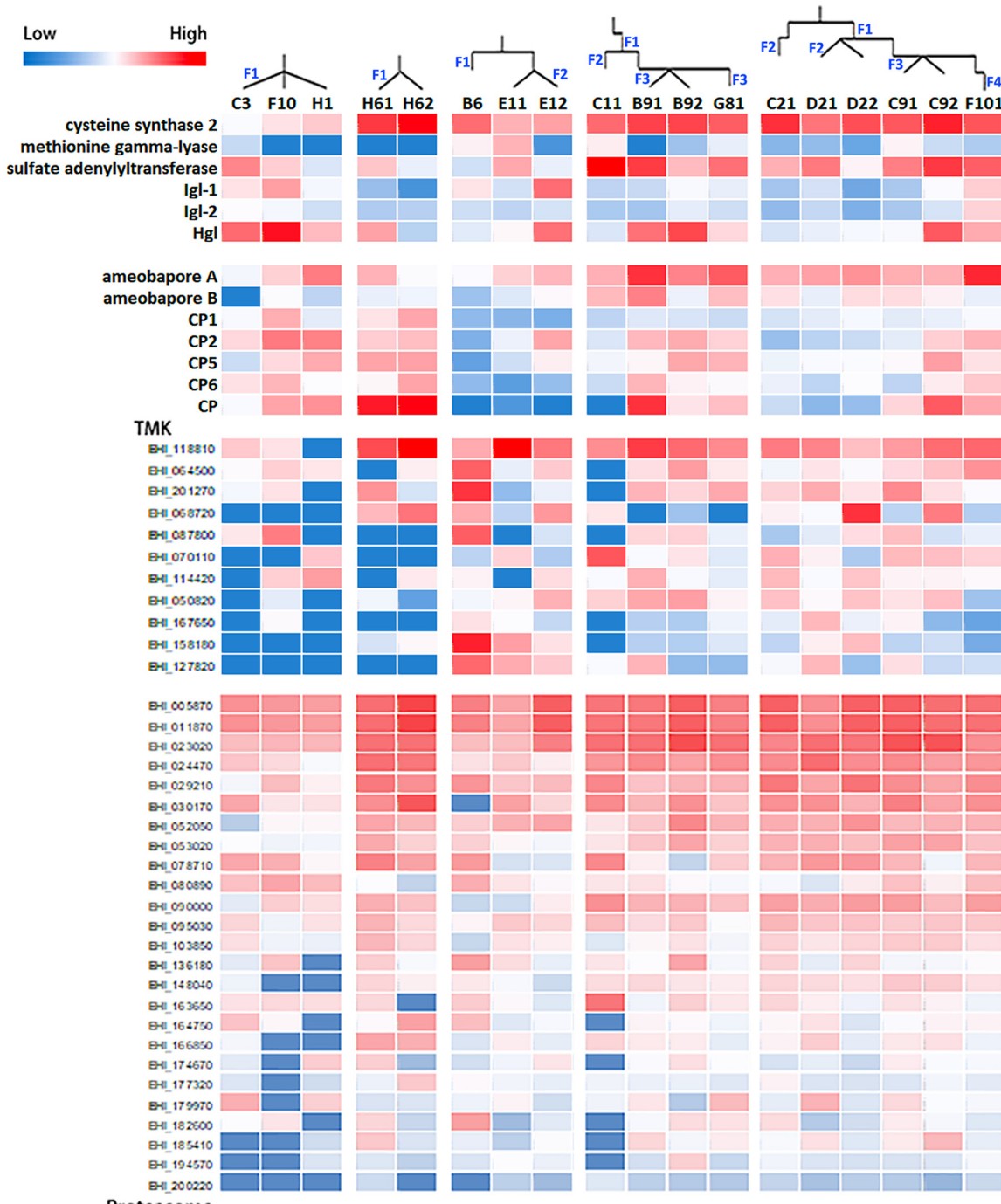

**FIG 9** Heat map of cysteine-related genes, CP, AP, TMK, and proteasome genes in single trophozoites of the CHO incubation group. C11 (F2), B91 and B92 (F2), and G81 (F3) were generated by one trophozoite. C21 (F2), D21 and D22 (F2), C91 and C92 (F3), and F101 (F4) were generated by one trophozoite. H1, F10, and C3 are the F1 generation of one trophozoite. H61 and H62 are the F1 generation of one trophozoite. B6 (F1) and E11 and E12 (F2) were generated by one trophozoite.

*vivo* trophozoites are affected by various external and host factors. Thus, trophozoites are under greater pressure, which forces changes in their transcriptional profiles to converge. Without the pressure of the internal environment, trophozoites cannot maintain sustained virulence *in vitro*. In addition, parasite damage results in colitis or extraintestinal abscesses. The outcome of amoebiasis is thought to be caused by

different genotypes of trophozoites (46, 47) but may also be due to the alterations in the trophozoite transcription profiles.

*E. histolytica* has a high demand for cysteine (10). Although amoebae have the ability to synthesize cysteine, cysteine is usually provided in the *in vitro* culture medium to meet its requirement (48, 49). The present study indicates that the trophozoites prepared reduced the expression of Igls under cysteine-deficient conditions. The upregulation of cysteine synthesis and the downregulation of Igls can occur *in vitro* but probably not *in vivo* due to the adequate supply of cysteine. Such adaptation and survival of the trophozoites under key nutrient-deficient environments reduced their own virulence simultaneously. Promoting trophozoite virulence is a complex process requiring the participation of multiple factors of the host, and increasing sulfur metabolism may be critical in this process, thereby determining the outcome of disease.

The present study identified the periodic variation in the transcriptional profiles of *E. histolytica* trophozoites. Multicellular transcriptome studies cover up these variations. Single-cell transcriptome studies reveal the special gene expression profile of trophozoites *in vitro*. This periodic variation in genes is reported in many other studies (50, 51) and usually exists with prolonged periods of infection, suggesting its involvement in the evasion of the immune system by these parasites. The mechanisms of persistence of these organisms are thought to be, in part, due to the change in the surface proteins. For example, *Plasmodium falciparum* has three families of var genes that are independently expressed. The highest variation rate of these families can reach 2% per generation (52, 53). Two species of flagellates are also reported with these kinds of changes at the transcriptional level. The intestinal protozoa *Giardia* harbors a family of variant surface glycoproteins (VSPs) with 100 to 150 members whose surface expression changes at a rate of one variation every 5 to 13 generations (54). The blood protozoan *Trypanosoma brucei* has more than thousands of VSPs that change at a rate of $10^{-2}$ to $10^{-7}$ variations per generation (55, 56). The results from this study indicate that TMK genes change at a certain rate of variations per generation. TMKs, as transmembrane proteins, may have similar immune escape functions to those of the mutant proteins mentioned above. However, the function of TMKs is not limited to immune evasion (17–20). Studies on TMKs indicate that this gene family regulates cell survival, proliferation, differentiation, metabolism, and migration to a considerable extent. The presence of TMKs in the plasma membrane offers one kind of potential mechanism to understand the strong ability of the organism to respond to the changing environment from the host. Thus, TMKs play critical roles as signal perceivers and transducers in higher eukaryotes (21–23). The single trophozoites from the CHO coincubation group displayed an upregulation of TMK genes distinct from that for the two other groups. Some TMKs were only highly expressed in the trophozoites from the CHO incubation group, suggesting their probable involvement in environmental perception and signal transmission. Some TMKs were expressed in almost any trophozoite, indicating that they can work in cell survival and proliferation. Other TMKs were expressed in a portion of trophozoites or were absent in this study, thereby remaining with unknown function. Results suggest that some TMK genes can play important roles in signal transmission under cysteine-deficient conditions and can regulate downstream molecules, such as the cell metabolism of trophozoites with certain regularity.

The present study involves the use of scRNA-seq to study the transcriptional profiles of trophozoite mRNAs *in vitro* and *in vivo*. Sequencing mRNAs at a single-trophozoite level enables the discovery of the transcriptions of trophozoites that are inconsistent or asynchronous *in vitro*. However, host factors force the *in vivo* trophozoites to change their transcriptional profiles so that they converge. The expression of the intermediate subunit of lectin of *E. histolytica* trophozoites is reduced under cysteine-deficient conditions and suggests the related regulation of CS and Igls. Furthermore, the periodic variation in the transcriptional profiles of *E. histolytica* trophozoites and for the TMK gene family is critical in regulating trophozoite proteasome metabolism. In contrast to bulk-cell sequencing, scRNA-seq can clarify the variation and differences in the

mSystems®

transcriptome. This finding is beneficial in understanding the biological characteristics and virulence variation of amoebic trophozoites.

## MATERIALS AND METHODS

**Trophozoites and cell culture.** The trophozoites of *E. histolytica* HM1:IMSS strains were grown under axenic conditions at 36.5°C in YIMDHA-S medium (49) containing 10% (vol/vol) heat-inactivated adult bovine serum. Parasites were grown for 72 h (log phase) for use in all experiments. Cysteine-deprived *E. histolytica* HM1:IMSS trophozoites were cultured under cysteine-deficient conditions by using YIMDHA-S medium without cysteine hydrochloride. CHO-K1 cells were cultured in Ham's F12 nutrient medium supplemented with 10% fetal bovine serum, 100 U/ml penicillin, and 100 $\mu$g/ml streptomycin. Cells were grown in an incubator maintained at 37°C with 5% $CO_2$. CHO-K1 cells ($10^4$) were cultured in 96-well plates (Costar, NY, USA) and incubated overnight to prepare the CHO cell monocyte layer. Then, a single trophozoite was added. Jurkat cells were cultured in RPMI 1640 medium supplemented with 10% fetal bovine serum, 100 U/ml penicillin, and 100 $\mu$g/ml streptomycin.

**Animal model for amoebic liver abscess.** Six-week-old male hamsters were obtained from Shanghai Songlian Experimental Animal Factory. An amoebic liver abscess (ALA) was induced by directly inoculating $1 \times 10^6$ axenic *E. histolytica* HM1:IMSS trophozoites into the liver as described previously (57). All animal experiments were performed in strict accordance with the guidelines from the Regulations for the Administration of Affairs Concerning Experimental Animals (1988.11.1) and approved by the Institutional Animal Care and Use Committees of our institutions (permit no. 20160225-097). All efforts were made to minimize suffering.

**Single trophozoite preparation. (i) Axenic culture group.** The trophozoites used in this study were from a clonal cultured single trophozoite. When trophozoites were grown to logarithmic growth phase, trophozoites were separated to single trophozoites using the limiting dilution method. The single trophozoites were transferred to blank 96-well plates in YIMDHA-S medium and collected for scRNA-seq.

**(ii) CHO incubation group.** Single trophozoites separated by the limiting dilution method were transferred into 96-well plates with CHO cell monocyte layers in YIMDHA-S medium. The plates were incubated at 37°C under anaerobic conditions. After the single trophozoites proliferated, the proliferated trophozoites were separated using the limiting dilution method. A portion of the separated single trophozoites was subjected to scRNA-seq. The remaining single trophozoites were added to new 96-well plates with CHO cell monocyte layers. These steps were repeated to collect multiple offspring from the original one trophozoite.

**(iii) *In vivo* group.** The clonal cultured single trophozoites were grown to a logarithmic growth phase for *in vivo* study. ALA was induced by directly inoculating $1 \times 10^6$ trophozoites into a hamster's liver. The hamsters were sacrificed 7 days later, and the livers were removed and placed in a sterile environment. The single trophozoites in the ALA tissue were separated using the limiting dilution method. These separated single trophozoites were subjected to scRNA-seq.

**Single-cell transcriptome analysis.** cDNA synthesis and library preparation of the single trophozoites were performed using the REPLI-g single-cell RNA library kit (Qiagen, Dusseldorf, Germany) in accordance with the manufacturer's recommendations. Briefly, $10^3$ trophozoites/ml were suspended in YIMDHA-S medium. Single trophozoites were separated using the limiting dilution method and added with water to bring the volume to 7 $\mu$l, and the solution was transferred to a microtube. After adding lysis buffer, the RNA from the single trophozoites was harvested and reverse transcribed into cDNA in accordance with the manufacturer's manual.

The transcript cDNA was preamplified using REPLI-g SensiPhi DNA polymerase and then sent for library preparation. The library preparation included end repair, A addition, adapter ligation, cleanup, and size selection of the amplified cDNA. The library was purified using Agencourt AMPure beads (Beckman Coulter, CA, USA) in accordance with the manufacturer's recommendations. After preparing high-diversity libraries, the samples were quantified. Single-trophozoite samples were pooled separately and loaded proportionally to their expected cell content for sequencing on an Illumina NextSeq 4000.

The resulting expression matrix of single-cell transcriptomes was used for clustering and analysis by using the Seurat package of scRNA-seq analysis tools (58, 59). For the pathway analysis, differential transcriptions were mapped to the terms in the KEGG database by using the KAAS program (http://www.genome.jp/kaas-bin/kaas_main).

**Quantitative real-time RT-PCR.** The *E. histolytica* strain HM1:IMSS trophozoites ($1 \times 10^5$ for each group) were grown in YIMDHA-S medium with or without 6 mM L-cysteine hydrochloride for 24, 48, and 60 h. The trophozoites were harvested, and the total RNA of these trophozoites was purified using the RNeasy Plus minikit (Qiagen, Dusseldorf, Germany). cDNA was synthesized using the PrimeScript first-strand cDNA synthesis kit (TaKaRa, Shiga, Japan) with oligo(dT) primers. The cDNA of the trophozoite was used for qRT-PCR. qRT-PCR was conducted in a final reaction volume of 20 $\mu$l in accordance with the manufacturer's recommendations on the ABI 7500 real-time PCR system (Applied Biosystems, CA, USA). Reactions were performed in 96-well plates with the SYBR Premix Ex *Taq* (TaKaRa), which contained primers for CS, MGL, Hgl, Igl-1, Igl-2, CP2, CP5, and AP-A genes of amoeba (see Table S4 in the supplemental material). The amplification cycling conditions were as follows: 30 s at 95°C and 40 cycles of 5 s at 95°C and 35 s at 60°C. qRT-PCR for gene expression for each cytokine was conducted during the log phase of product accumulation, during which the threshold cycle ($C_T$) values correlated linearly with the relative DNA copy numbers. Each experiment was performed at least thrice.

**Amoebic phagocytic and cytotoxic assays.** Jurkat cells were grown and maintained in RPMI 1640 medium, whereas the *E. histolytica* strain HM1:IMSS trophozoites were grown in YIMDHA-S medium with

or without 6 mM L-cysteine hydrochloride for 60 h. For the phagocytic assay, trophozoites were labeled with $2 \mu M$ CFSE for 5 min at room temperature. Jurkat cells were incubated with $5 \mu M$ DiD at 55°C for 20 min to label the cells and induce cell death. Normal or cysteine-deprived trophozoites were coincubated with DiD-labeled heat-killed Jurkat cells (1:10 ratio) at 37°C for 5 or 10 min. Trophozoites and cells were fixed with 4% paraformaldehyde, and flow cytometry was performed using an ImageStreamX Mark II. For the cytotoxic assay, normal or cysteine-deprived trophozoites were coincubated with living Jurkat cells (1:20 ratio) at 37°C for 10 or 20 min. The cells were stained with PI, and flow cytometry was performed as described previously. The experiments were repeated thrice.

**Fluorescence imaging of trophozoites and cells.** Trophozoites and Jurkat cells from amoebic phagocytic assays were also used for confocal microscopy. Cell suspensions containing CFSE-labeled trophozoites and DiD-labeled Jurkat cells were placed onto glass slides, mounted with coverslips, and examined using a Leica TCS SP8 microscope. PI-stained Jurkat cells from amoebic cytotoxic assays were also imaged using fluorescence microscopy.

**iTRAQ proteomic analysis.** The axenic-cultured trophozoites were collected directly or incubated with CHO cells at a ratio of 1:2 for 2, 4, and 6 h. The proteins of trophozoites were isolated and quantified using the Bradford method (Bio-Rad, CA, USA). The samples were then reacted as previously described (60). Briefly, each sample was digested and labeled using an 8-plex iTRAQ labeling kit (Applied Biosystems). The iTRAQ-labeled samples were fractionated through two-dimensional liquid-phase chromatography and nanoscale high-performance liquid chromatography. Data were acquired automatically and analyzed using the Mascot version 2.3.02 and the Scaffold version 4.3.2. A fold change cutoff ratio of >1.2 or <0.833 was selected to designate differentially expressed proteins (61). The GO program Blast2GO (BioBam, Valencia, Spain) was used to annotate differential expression proteins to create the histograms of GO annotation, including CC, BP, and MF.

**Data availability.** The raw scRNA-seq data are available at NCBI BioProject database under BioProject accession number PRJNA680388. The raw iTRAQ data are available at the jPOST database under accession number JPST001018 (PXD022686). The assembled RNA sequencing data and proteomics data are available in github (https://github.com/MengFeng-Fudan/Data-msystems2020).

## SUPPLEMENTAL MATERIAL

Supplemental material is available online only.

**FIG S1**, TIF file, 0.3 MB.
**FIG S2**, TIF file, 2.3 MB.
**FIG S3**, TIF file, 0.7 MB.
**FIG S4**, TIF file, 2.9 MB.
**FIG S5**, TIF file, 0.3 MB.
**TABLE S1**, DOCX file, 0.1 MB.
**TABLE S2**, DOCX file, 0.1 MB.
**TABLE S3**, XLSX file, 0.1 MB.
**TABLE S4**, DOCX file, 0.1 MB.
**TABLE S5**, XLSX file, 0.1 MB.

## ACKNOWLEDGMENTS

We thank Qiao Wang at the Department of Medical Microbiology and Parasitology, School of Basic Medical Sciences, Fudan University, for assisting in data analysis.

This work was supported by National Natural Science Foundation of China (81630057) and the National Key Research and Development Program of China (2018YFA0507304).

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
