## [Reviewer comments · mSystems]

Single-cell RNA-sequence reveals that the switching of the transcriptional profiles of cysteine-related genes alter the virulence of *Entamoeba histolytica*

Meng Feng, Yuhan Zhang, Hang Zhou, Xia Li, Yongfeng Fu, Hiroshi Tachibana, and Xunjia Cheng

Corresponding Author(s): Xunjia Cheng, School of Basic Medical Sciences, Fudan University

Review Timeline:

Submission Date:	October 22, 2020
Editorial Decision:	November 5, 2020
Revision Received:	November 9, 2020
Editorial Decision:	November 11, 2020
Revision Received:	November 24, 2020
Accepted:	November 24, 2020

Editor: Marta Gaglia

Reviewer(s): Disclosure of reviewer identity is with reference to reviewer comments included in decision letter(s). The following individuals involved in review of your submission have agreed to reveal their identity: Mario Alberto Rodríguez (Reviewer #1)

Transaction Report:

DOI: <https://doi.org/10.1128/mSystems.01095-20>

Response to Reviewer 1:

Thank you very much for your valuable comments.

However, to separate single trophozoites from axenic cultures and to inoculate the animals you started from cultures in logarithmic growth-phase, whereas, according with Materials and methods, for trophozoites incubated with CHO cells you initiated with a single trophozoite over the cells and after proliferation (-how long?) you isolated single cells. Thus, probably transcriptional changes could be also due to different growth phases. To be comparable you should incubate trophozoites from a culture in exponential growth with cells in a relation of 1:1 to 1: 10 for a determined time (may be 15 to 60 min), and then isolate single cells.

Reply: We revised the manuscript and redrawn figure 1A to clearly describe the steps of separation of single trophozoite. The trophozoites used in this study were from a clonal cultured single trophozoite. When trophozoites were grown in logarithmic growth-phase, partial of the trophozoites were inoculated into the animals. Remained trophozoites were separated to single trophozoite by limiting dilution method, then transferred to new 96 well plates with or without CHO cells monolayer in YIMDHA-S medium. The experimental conditions of axenic culture group and CHO incubation group were consistent, so the experimental results of the two groups were comparable.

Changes in the revised highlighted manuscript: Page 19 lines 394 to 401, Page 20 lines 407 to 408, Figure 1A.

Minor concerns:

1) Lines 141-142: you have to mention which type of genes are included in the different groups of expression profiles (BP, CC, MF)

Reply: We added the information in supplementary data.

Changes in the revised highlighted manuscript and supplementary data: Page 7 line 141.

2) Lines 175-183: you mentioned that the expression level of CS remained high in

cysteine-deficient cultures, but the expression levels of Igl-1, and Igl-2 decreased in the same condition. So, why you declared that there is positive correlations between CS and Igl-1 and between CS and Igl-2?. In addition is not clear the mechanism that guarantee the expression of Hgl under this condition

Reply: We revised the words in this sentence. There was negative correlation between CS and Igl-1 and between CS and Igl-2. We didn't study the regulation mechanism of expression of Hgl. We deleted related statements.

Changes in the revised highlighted manuscript: Page 9 line 177.

3) Line 233: TKMs are not proteases.

Reply: We have revised to protein kinase as suggestion.

Changes in the revised highlighted manuscript: Page 11 line 227.

4) Figure 5: You must to add in the figure legend what do mean c and w/o (for example 24 h c; 24 h w/o)

Reply: We added information in figure legend. "c" mean medium contained normal concentration of cysteine. "w/o" mean medium without cysteine.

Changes in the revised highlighted manuscript: Figure 5 legend.

5) Fig 6. What condition represents each normalized counts in the graphs?

Reply: Data of normalized counts used in figure 6 were software corrected counts to compare the genes' expression levels of different single trophozoites.

Response to Reviewer 2:

Thank you very much for your valuable comments.

1) It is not clear if all of the work presented was derived from a single trophozoite clone. If it has not, then there is an issue about the degree to which the transcriptional changes are selected by the different in vitro and in vivo conditions vs are random results from selection of different subpopulations within the amebic culture.

Reply: We revised the manuscript and redrawn figure 1A to clearly describe the steps of separation of single trophozoite. The trophozoites used in this study were from a clonal cultured single trophozoite. When trophozoites were grown in logarithmic growth-phase, partial of the trophozoites were inoculated into the animals. Remained trophozoites were separated to single trophozoite by limiting dilution method, then transferred to new 96 well plates with or without CHO cells monolayer in YIMDHA-S medium. The experimental conditions of axenic culture group and CHO incubation group were consistent, so the experimental results of the two groups were comparable.

Changes in the revised highlighted manuscript: Page 19 lines 394 to 401, Page 20 lines 407 to 408, Figure 1A.

2) Transcriptome was most different in the CHO cell co-culture. An important control is growth of the trophozoites in the medium from the CHO cell co-incubation since this is very different from the parasite culture medium

Reply: Single trophozoite was added to CHO cells monolayer in YIMDHA-S medium. We revised the manuscript to clearly describe the medium used in the study.

Changes in the revised highlighted manuscript: Page 19 lines 400 to 401.

November 5, 2020

Prof. Xunjia Cheng
School of Basic Medical Sciences Fudan University
Medical Microbiology & Parasitology
No. 138 Yixueyuan Road
Shanghai 200032
China

Re: mSystems01095-20 (Single-cell RNA-sequence reveals that the switching of the transcriptional profiles of cysteine-related genes alter the virulence of *Entamoeba histolytica*)

Dear Prof. Xunjia Cheng:

Below you will find the comments of the reviewers. As you will see the reviewers were satisfied with your responses to their previous comments. However, prior to acceptance you will need to ensure that the RNA sequencing and proteomics data you collected are made publicly available and to add a data availability statement to the Materials and Methods section reflecting this, as per mSystems open data policy (<https://journals.asm.org/content/open-data-policy>).

To submit your modified manuscript, log onto the eJP submission site at <https://msystems.msubmit.net/cgi-bin/main.plex>. If you cannot remember your password, click the "Can't remember your password?" link and follow the instructions on the screen. Go to Author Tasks and click the appropriate manuscript title to begin the resubmission process. The information that you entered when you first submitted the paper will be displayed. Please update the information as necessary. Provide (1) point-by-point responses to the issues raised by the reviewers or editor as file type "Response to Reviewers," not in your cover letter, and (2) a PDF file that indicates the changes from the original submission (by highlighting or underlining the changes) as file type "Marked Up Manuscript - For Review Only."

Due to the SARS-CoV-2 pandemic, our typical 60 day deadline for revisions will not be applied. I hope that you will be able to submit a revised manuscript soon, but want to reassure you that the journal will be flexible in terms of timing, particularly if experimental revisions are needed. When you are ready to resubmit, please know that our staff and Editors are working remotely and handling submissions without delay. If you do not wish to modify the manuscript and prefer to submit it to another journal, please notify me of your decision immediately so that the manuscript may be formally withdrawn from consideration by mSystems.

Sincerely,

Marta Gaglia

Editor, mSystems

Journals Department
Reviewer comments:

Reviewer #1 (Comments for the Author):

I would like to thank the authors on addressing the suggested changes and answering my questions.

Reviewer #2 (Comments for the Author):

the authors have responded comprehensively to the concerns raised in the prior review

Response to Editor:

Thank you very much for your kind comments.

However, prior to acceptance you will need to ensure that the RNA sequencing and proteomics data you collected are made publicly available and to add a data availability statement to the Materials and Methods section reflecting this,

Reply: The RNA sequencing data and proteomics data are now available in github (<https://github.com/MengFeng-Fudan/Data-msystems2020>). We added a data availability statement to the Materials and Methods section.

Changes in the revised highlighted manuscript: Page 23 lines 485 to 487.

Response to Reviewer 1:

Thank you very much.

Response to Reviewer 2:

Thank you very much.

November 11, 2020

Prof. Xunjia Cheng
School of Basic Medical Sciences Fudan University
Medical Microbiology & Parasitology
No. 138 Yixueyuan Road
Shanghai 200032
China

Re: mSystems01095-20R1 (Single-cell RNA-sequence reveals that the switching of the transcriptional profiles of cysteine-related genes alter the virulence of *Entamoeba histolytica*)

Dear Prof. Xunjia Cheng:

Thank you for your edits and depositing the processed data. Please also make raw reads for your sequencing available, as per our data policy, and add the accession number to the manuscript. Consult <https://journals.asm.org/content/open-data-policy> for more guidance.

To submit your modified manuscript, log onto the eJP submission site at <https://msystems.msubmit.net/cgi-bin/main.plex>. If you cannot remember your password, click the "Can't remember your password?" link and follow the instructions on the screen. Go to Author Tasks and click the appropriate manuscript title to begin the resubmission process. The information that you entered when you first submitted the paper will be displayed. Please update the information as necessary. Provide (1) point-by-point responses to the issues raised by the reviewers as file type "Response to Reviewers," not in your cover letter, and (2) a PDF file that indicates the changes from the original submission (by highlighting or underlining the changes) as file type "Marked Up Manuscript - For Review Only."

Due to the SARS-CoV-2 pandemic, our typical 60 day deadline for revisions will not be applied. I hope that you will be able to submit a revised manuscript soon, but want to reassure you that the journal will be flexible in terms of timing, particularly if experimental revisions are needed. When you are ready to resubmit, please know that our staff and Editors are working remotely and handling submissions without delay. If you do not wish to modify the manuscript and prefer to submit it to another journal, please notify me of your decision immediately so that the manuscript may be formally withdrawn from consideration by mSystems.

Sincerely,

Marta Gaglia

Editor, mSystems

Journals Department
Response to Editor:

Thank you very much for your kind comments.

Thank you for your edits and depositing the processed data. Please also make raw reads for your sequencing available, as per our data policy, and add the accession number to the manuscript.

Reply: The raw scRNA-seq data are available at NCBI BioProject database under BioProject accession number PRJNA680388. The raw iTARQ data are available at jPOST database under accession number JPST001018 (PXD022686). The assembled RNA sequencing data and proteomics data are available in github (<https://github.com/MengFeng-Fudan/Data-msystems2020>). We added a data availability statement to the Materials and Methods section.

We finished the upload of all the data and announced immediately. It is currently being processed and may take a few days.

Changes in the revised highlighted manuscript: Page 23 lines 485 to 490.

November 24, 2020

Prof. Xunjia Cheng
School of Basic Medical Sciences Fudan University
Medical Microbiology & Parasitology
No. 138 Yixueyuan Road
Shanghai 200032
China

Re: mSystems01095-20R2 (Single-cell RNA-sequence reveals that the switching of the transcriptional profiles of cysteine-related genes alter the virulence of *Entamoeba histolytica*)

Dear Prof. Xunjia Cheng:

Thank you for depositing the omics data.

Your manuscript has been accepted, and I am forwarding it to the ASM Journals Department for publication. For your reference, ASM Journals' address is given below. Before it can be scheduled for publication, your manuscript will be checked by the mSystems senior production editor, Ellie Ghatineh, to make sure that all elements meet the technical requirements for publication. She will contact you if anything needs to be revised before copyediting and production can begin. Otherwise, you will be notified when your proofs are ready to be viewed.

Sincerely,

Marta Gaglia
Editor, mSystems

Journals Department
Table S1: Accept
Fig. S4: Accept
Fig. S1: Accept
Fig. S5: Accept
Table S4: Accept
Table S5: Accept
Table S2: Accept
Fig. S2: Accept
Fig. S3: Accept
Table S3: Accept